# Beyond the Bark: An Overview of the Chemistry and Biological Activities of Selected Bark Essential Oils [note 1]

**DOI:** 10.3390/molecules27217295

**Published:** 2022-10-27

**Authors:** Melanie Graf, Iris Stappen

**Affiliations:** Department of Pharmaceutical Sciences, University of Vienna, 1090 Vienna, Austria

**Keywords:** Cinnamomum verum, Cedrelopsis grevei, Drypetes gossweileri, Cryptocarya massoy, Vanillosmopsis arborea, Cedrus deodara, bark, essential oils

## Abstract

Essential oils have been used by indigenous peoples for medicinal purposes since ancient times. Their easy availability played an important role. Even today, essential oils are used in various fields—be it as aromatic substances in the food industry, as an aid in antibiotic therapy, in aromatherapy, in various household products or in cosmetics. The benefits they bring to the body and health are proven by many sources. Due to their complex composition, they offer properties that will be used more and more in the future. Synergistic effects of various components in an essential oil are also part of the reason for their effectiveness. Infectious diseases will always recur, so it is important to find active ingredients for different therapies or new research approaches. Essential oils extracted from the bark of trees have not been researched as extensively as from other plant components. Therefore, this review will focus on bringing together previous research on selected bark oils to provide an overview of barks that are economically, medicinally, and ethnopharmaceutically relevant. The bark oils described are *Cinnamomum verum*, *Cedrelopsis grevei*, *Drypetes gossweileri*, *Cryptocarya massoy*, *Vanillosmopsis arborea* and *Cedrus deodara*. Literature from various databases, such as Scifinder, Scopus, Google Scholar, and PubMed, among others, were used.

## 1. Introduction

Essential oils (EO) are found in various parts of plants like leaves, roots, seeds, fruits, as well as bark and wood. They are secondary metabolites and therefore may occur in some species or parts of the plant and not or less in others. Depending on the location or the soil properties the plant grows in and depending on the species, the plant produces EOs with different composition and properties. [1,2] Chemically, EOs mainly consist of mono- and sesquiterpenoid hydrocarbons as well as phenylpropanoids that are biosynthesized in the plants via phosphoenolpyruvate. From this key building block, terpenoids are formed by the mevalonic acid- or the methyl erythritol phosphate pathways. Their basic structural element is the isoprene unit; monoterpenoid hydrocarbons comprising of two, sesquiterpenoid hydrocarbons of three units. Phenylpropanoids are derived by the shikimic acid pathway, representing the aromatic fraction of the EO. These main structures further can be oxygenated to alcohols, ethers, aldehydes, ketones, and acetates. [2,3] Plants produce these highly volatile, lipophilic, and very complex multi-component mixtures as a defense mechanism against microorganisms, predators, and pests, but also to attract pollinators [1]. In most cases, one to four main compounds make up to 90% of the oil, whereas the concentration of the other components are significantly lower [3]. Extraction methods for obtaining EO have evolved considerably in recent years. Several new extraction methods have significantly improved the quality, purity and yield of the oils and facilitated their analysis [2]. But they also played a major role in earlier times. There are records of the Persians using simple equipment for hydrodistillation as early as 3500 BC, which underscores the importance of these EOs for centuries. Especially in the context of traditional use, herbal medicine is more accessible to most people than conventional medicine. EOs are used against a wide range of diseases and are particularly appreciated for their multiple properties. As for the various bark oils and their main components to be presented here, these properties include antimicrobial, insecticidal, anti-tyrosinase, and antioxidant activities. However, often not only one main component of the EO is involved in the mechanism of action, but there is a synergistic effect between several components of the oil. EOs also play an increasingly important role in modern medicine. In view of the increasing resistance to antibiotics and emerging infectious diseases, EOs appear to have great potential for future applications and development in terms of their antibiotic and also efficacy-enhancing properties [4]. However, it is not only the pharmacological sector that is interested in the study of EOs. The anti-inflammatory properties and non-toxicity that some EOs exhibit are also used in cosmetics. In the cosmetic and perfume industry, EOs additionally play an important role due to their unique odor [5]. The food industry also tries more and more to limit the use of conventional antibiotics and replace them with natural substances. The approach of using EOs as additives seems to bring good results. Agriculture also benefits from their repellent or larvicidal activity as new insecticides [6]. The various healing properties of these oils have already been used by indigenous peoples and are increasingly being explored for other purposes. In the course of this review, bark EOs of selected trees have been viewed regarding their chemistry, their different properties and fields of application, as well as their prospect to the future medical development. These trees are: *Cinnamomum verum*, *Cedrelopsis grevei*, *Drypetes gossweileri*, *Cryptocarya massoy*, *Vanillosmopsis arborea* and *Cedrus deodara*. They were chosen for their economic but also for their ethnopharmaceutical importance as well as their unique chemical composition. An overview of their main activities described in this review is listed in Table 1. Moreover, their main constituents—listed in Figure 1—as well as the EOs compositions regarding their variations are discussed.

## 2. Essential Oils of Selected Barks

### 2.1. Cinnamomum verum J. Presl (Lauraceae)

*C. verum*, also known as *C. zeylanicum* Blume, is an evergreen tree found in Asia and Australia that can reach a height of 20 m [11,35]. The trees are characterized by leathery, ovate leaves that have three prominent veins. They turn from reddish to dark green depending on age. The inflorescence of pale-yellow flowers is axillary or in terminal panicles. The fruit is a dark purple to black colored ovoid drupe. During rainy periods the 3–4-year-old stems are cut off to make the bark easier to peel off. The outer skin of the bark is removed, and the remaining bark is rubbed off with a brass block. A special round knife is then used to obtain the compound quills of the cinnamon. The quills are dried and then disinfected with sulfur dioxide, which also is the reason for the golden hue of the quills [35]. The genus *Cinnamomum* includes about 250 species that grow mainly in Asia and Australia. The main cultivation of *C. verum* takes place in its country of origin, Sri Lanka [36]. From there, cinnamon is exported to various countries around the world [11]. Records show that exportation began as early as the 16th and 17th centuries. The uses of cinnamon seem almost limitless; for centuries it has been a valued spice. It is also used in Ayurvedic medicine and was used for mummification in ancient Egypt. The biological properties of the *Cinnamomum* genus are enormous. Several studies show its antimicrobial, insecticidal, acaricidal, antityrosinase, antimycotic, anticancer, anti-inflammatory, antidiabetic, antioxidant and antimutagenic activities [8,37]. *C. verum* is also used as scent for perfumes [9].

A GC-MS analysis of hydrodistilled *C. zeylanicum* bark purchased in Turkey identified nine different compounds of the EO: (E)-cinnamaldehyde (68.95%; Figure 1a), benzaldehyde (9.94%), (E)- cinnamyl acetate (7.44%; Figure 1b), limonene (4.42%), eugenol (2.77% Figure 1c), α-pinene (1.64%; Figure 1d), 1,8-cineol (1.55%), linalool (1.38%; Figure 1e) and (E)-cinnamic acid (1.15%). In particular, the major constituent (E)-cinnamaldehyde showed different variations with respect to its total content in the EO. Depending on where the plant material was collected, the content of the individual components of the oil also differed [8].

Two different studies analyzed plant material of *C. zeylanicum* bark collected from different parts in Turkey. Nine different compounds were isolated in one study and 22 compounds were isolated in the other study. In both GC-MS analyses (E)- cinnamaldehyde and (E)-cinnamyl acetate were among the main compounds [8,10].

The hydrodistilled *Cinnamomum* EO showed anticoagulant effects, while the aqueous extract of the same bark showed no reduction in clotting time in a blood test, suggesting that the active ingredient seems to be a component of the EO. The authors could not fully determine exactly which component was responsible for this biological effect, due to the complexity of the oil [7].

To evaluate antimicrobial activity, Unlu et al. [8] tested the EO against 21 bacteria and four *Candida* species. Screening against the bacteria and yeasts of interest for their minimum inhibitory concentration (MIC) using the disc diffusion method (DDM) showed the high efficacy against Gram-positive (*Staphylococcus, Streptococcus, Enterococcus*) and Gram-negative (*Pseudomonas aeruginosa*) bacteria as well as against different *Candida* strains. The components responsible for the antibacterial property were identified as cinnamaldehyde and eugenol, which are both phenylpropanoids and known for their high antimicrobial potential [3]. They either damage the cell walls of the bacteria or inhibit an essential enzyme—or even both [8]. Cinnamaldehyde is also used in periodontal disease. Its antimicrobial activity has been demonstrated in many studies. High biological activities are attributed to this component throughout the genus *Cinnamomum*, as reviewed by Wang and coworkers [38].

In the context of antidiabetic activity, results are biased, some studies reporting stronger inhibition of α-amylase by the EO, others weaker. Interestingly, the inhibitory effect of the whole oil on this enzyme was 15.25%, whereas the isolated substances (E)-cinnamaldehyde and (E)-cinnamyl acetate exhibited an α-amylase-inhibition of 21.30% and 7.29%, respectively. Different ratios of (E)-cinnamyl acetate to (E)-cinnamaldehyde were tested. The inhibition of the enzyme increased by up to 38.74%. This was the highest measured α-amylase inhibitory activity obtained in this study by different mixtures of the individual components of the EO [10].

In the field of Alzheimer’s disease, in a study by Tepe and Ozaslan, the oil and both main compounds of the oil showed comparable results in the inhibition of MAO-A and MAO-B with rasagiline, as well as a 99.0 % inhibitory effect on cholinesterases. The EO and (E)-cinnamaldehyde therefore seem to have great potential for the future treatment of Alzheimer’s disease [10]. Natural antioxidants play an important role in nutrition and are believed to prevent many diseases, including cancer. Additionally, they are used in the food industry to limit food spoilage [37]. An in vitro study of antioxidant properties showed inhibition of 3-nitrotyrosine formation by the EO and eugenol with IC_50_ values of 18.4 µg/mL and 46.7 µM/mL, respectively. In contrast, (E)-cinnamaldehyde and linalool showed no activity in inhibiting peroxynitrite-induced nitration and lipid peroxidation [11].

The neuroprotective activity of *C. zeylanicum* EO was also demonstrated in an in vitro model testing the viability of PC12 cells and reactive oxygen species after exposure to 6- hydroxydopamine. Pretreatment of PC12 cells with the EO and its main compound cinnamaldehyde reduced ROS and cytochrome C and increased survivin. This demonstrated efficacy against the cytotoxicity of 6-OHDA by inhibiting the p44/42 pathway in PC12 cells suggesting future application in the treatment of Parkinson’s disease. [12]

An MTT (3-(4,5-Dimethylthiazol-2-yl)-2,5-diphenyltetrazoliumbromid) assay was prepared to access the EOs’ future use in tumor therapy. A dose above 15 µg/mL of EO was able to show cytotoxicity against cell lines 5RP7 (H-ras active-rat fibroblasts) and F2408 (normal rat fibroblasts). The time-dependent cytotoxicity towards H-ras-active cells, which was also found, could probably indicate an application in ras-oncogene-induced carcinomas, which has also been demonstrated for monoterpenes such as limonene. [8,39] However, further studies need to be conducted to confirm this activity. The authors of the study also described that low concentrations of the oil induced controlled cell death and high doses induced necrotic death [8].

Further investigations tested the anti-inflammatory effects and tissue remodeling of *C. verum* EO. Biomarker proteins for inflammatory processes such as vascular cell adhesion molecule-1 (VCAM-1) and intercellular cell adhesion molecule-1, as well as epidermal growth factor receptor (EGFR), matrix metalloproteinase 1, and plasminogen activator inhibitor-1 could be inhibited by the EO. An immunomodulatory effect on macrophage colony-stimulating factor was also noted. Some signaling pathways that trigger inflammatory responses could be inhibited. Although many studies sound very promising, further studies are needed to determine clinical application and safety [13].

### 2.2. Cedrelopsis grevei Baill. (Ptaeroxylaceae)

*C. grevei* is a tree native to Madagascar, which is one of the most species-rich regions of the world. Malagasy people turned to natural medicine for a variety of reasons. One is because of the high cost of drugs in modern medicine, whereas another is a social aspect. As a result, minor injuries and illnesses tend to be healed by community elders and major illnesses are then taken care of by traditional healers, some of whom rely on the local flora. According to research on the most commonly used medicinal plants in Madagascar, *C. grevei* is one of the species with the greatest utility value [40]. The EO extracted from the bark of the tree, which grows up to 15 m high is also known as “katrafay”. Samples from different studies showed great variability in terms of EO composition. In one of their publications, Rakotobe and coworkers examined the hydrodistillation of 21 different bark samples from six different regions in Madagascar. They concluded that at least one of the samples contained 71 volatiles, eleven of which totaled about 10% of the oil. The most important compounds besides (E)-β-caryophyllene (Figure 1f) and caryophyllene oxide were monoterpene hydrocarbon α-pinene, sesquiterpene hydrocarbons α-copaene, ishwarane, selinenes, cadinenes, and oxygenated sesquiterpenes copaborneol, T-muurolol, α-cadinol, α- and γ-eudesmols, and α-bisabolol (Figure 1g). The difference between these results and another analysis of bark oil was that no *cis*-sesquisabine hydrate was found in that case. Rakotobe et al. also concluded that there are four distinct chemotypes that show their variability in EO composition probably due to differences in native soils [41]. Gauvin et al. analyzed the EO derived by steam distillation of the bark by GC-MS and found β-pinene (17.1%), *cis*-sesquisabinene hydrate (12.8%), caryophyllene oxide (7.0%) and δ-3-carene (4.2%) as main compounds [42].

Already in 2003, in the first report on the composition of the bark oil of *C. grevei* by Cavalli et al., the variability of this EO was noted. In one sample, (E)-β-caryophyllene (9.3%), α-copaene (7.7%), α-selinene (5.8%), δ-cadinene (4.9%), β-selinene (4.5%), α-humulene (3.3%), and β-bisabolene (2.8%) were detected. Over 114 compounds were identified. The main components ishwaren and (E)-β-caryophyllene were also found in other samples. Overall, all studies confirm the main occurrence of sesquiterpene hydrocarbons in the bark oil of *C. grevei* [43].

The EO was described by a perfumer from France as an amber oil with the odor of terpene hydrocarbons with woody, spicy, green, chypre and oriental notes [42]. The Malagasy use the bark of the trunk for various ailments. These include muscle fatigue and reduction of capillary fragility. It is also used as an ingredient in a cough syrup against persistent catarrh or in anti-hypertensive beverages according to the Madagascan pharmacopoeia [14,44,45]. Aphrodisiac properties have also been described [40]. The bark is further used for rheumatism or pulmonary and cardiovascular diseases, as well as a febrifuge. In addition, antifungal and antibiotic properties were reported [14].

It seems that the widespread benefits of the EO can be attributed to the terpene composition it contains. To conduct further studies on the biological activities, Tardugno et al. [14] performed a DPPH (2,2- diphenyl-1-picrylhydrazyl) assay and an ABTS+ spectrophotometric assay (2,2′- azino-bis (3-ethylbenzothiazoline-6-sulfonic acid)). For this purpose, the EO was divided into three different fractions. The entire EO was also examined. *C. grevei* EO showed a percentage inhibition of the radical scavenger of 4.65 ± 0.03, while Trolox as a positive control showed an inhibition of 54.10 ± 0.22%. The ABTS activity of the EO showed an inhibition of 19.41 ± 1.22% compared to the control of 98.40 ± 4.10%. The fraction containing the oxygenated sesquiterpenoids as components indicated a particularly significant increase in antioxidant activity: 4.78 ± 0.13 % in the DPPH and 88.56 ± 4.38 % in the ABTS+ assay [14].

Inhibition of microbial growth was determined using various human bacterial strains, yeasts, and dermatophytes. A closer look at the inhibition of bacterial growth by *C. grevei* EO revealed differences depending on the bacterial strain used. For *Escherichia coli* and *P. aeruginosa* it showed MIC values of >1.90 mg/L, for *Staphylococcus aureus* and *S. epidermidis* MIC values of 0.47 mg/L, respectively, and for *E. faecalis* a MIC of 0.95 mg/L was found. As a positive control the antibiotic chloramphenicol was used, which showed MIC values of 6.2 mg/L for *P. aeruginosa*, 25 mg/L for *E. coli* and 3.1 mg/L for *S. epidermidis* and *S. aureus*, respectively. *Thymus vulgaris* L. (*Lamiaceae*) EO, as a second positive control, demonstrated MIC values reaching from 0.5 to >2.00 mg/L. Thus, *C. grevei* EO showed a higher antibacterial activity compared to positive controls. In contrast to these findings, the EO of *C. grevei* did not indicate efficient growth inhibition on *C. albicans* as yeast and dermatophytes such as *Micosporum gypseum*, *Nannizzia gypsea*, *Tricophyton mentagrophytes* and *T. tonsurans*. However, the fraction of oxygenated sesquiterpenoids delivered selective activity against Gram-negative bacteria. Tardugno et al. concluded that this could be due either to the high percentage of oxygenated mono- and sesquiterpenoidss or also to a possible synergistic effect of the compounds in this fraction, which contained mainly sesquiterpene-alcohols palustrol, ledol, elemol and cubenol, as well as monoterpene-alcohol α-terpineol [14].

Finally, an MTT assay was performed to determine the effect on viability of A549 and CaCo-2 cancer cell lines. The EO showed an IC_50_ (μg/mL) value of 27.08 ± 1.90 against the A549 cancer cells and 54.06 ± 3.12 μg/mL against the CaCo2 cell line. Since the IC_50_ value (μg/mL) for A549 cancer cells was less than 30, the cytostatic property of this EO may have more practical uses and applications in the future. However, more research is needed for this purpose [14].

### 2.3. Drypetes gossweileri S. Moore (Euphorbiaceae)

*D. gossweileri*, also known locally as the “horseradish tree”, is a tree common in the Democratic Republic of Congo, Gabon, Equatorial Guinea, the Central African Republic and Cameroon. This tree plays an important role in traditional medicine of the mentioned countries [46]. It can reach a height of up to 42 m. The bark has a grey color and a pungent smell of horseradish [47].

Traditionally, the bark is used in many ways: as an analgesic, antipyretic and vermifuge; as a genital stimulant and depressant, but also against various pains, such as headaches, rheumatism, and toothache. It is also reported that the bark is applied against malaria, gonorrhoea and typhoid fever. Additionally, it is used in poison-fishing [46].

A GC-MS analysis of the EO of the bark by Agnaniet et al. identified eleven different components. During the distillation period of six hours, the oil was analyzed at different time points. After the six-hours distillation time, a large amount of decomposition products of glucosinolates was detected. Benzyl cyanide with 19.4% at 0.5 h and 73.7% after 6 h distillation was probably formed by the reaction between the cyanide ions and benzyl isothiocyanate (72.0% at 0.5 h and reached 5.2% at 6 h) [47].

In addition to benzyl isothiocyanate (Figure 1h) and benzyl cyanide (Figure 1i), the components in the bark oil were benzaldehyde (0.8%), benzyl alcohol (10.8%), benzyl formate (0.6%), linalool (0.6%), benzyl acetate (0.7%), β-caryophyllene (0.3%), caryophyllene oxide (0.3%), benzyl benzoate (0.2%), and benzyl salicylate (1.1%). The numbers indicate the percentage after six hours distillation and account for 93.4%. According to the result from GC-MS, the terpenes made up only a small part of the EO. Agnaniet et al. stated that the high content of benzyl alcohol probably showed that it emerged from benzyl isothiocyanate and not from the hydrolysis of the glucosinolates [47].

An EO analysis of horseradish bark harvested in Hawae and Ngomedzap (central region of Cameroon), with an EO extraction yield of 0.04% and an identified total content of 99.70%, showed the major constituent benzyl isothiocyanate in an amount of 63.19%, instead of benzyl cyanide, which accounted for 35.72% in that study. This differed from the study of Agnaniet et al. but can be explained by regional differences since this plant material was collected in the Nkomo-Moanda region [18,47]. In another study examining the main constituents of EO obtained from bark in Gabon and in the Central African Republic, benzyl isothiocyanate was the major compound in both reports. The EO of plants harvested in Gabon contained 1.3% benzaldehyde, 42% benzyl cyanide and 56.5% benzyl isothiocyanate. The plant material from the Central African Republic contained 0.2% benzaldehyde, 5.7% benzyl cyanide and 93.9% benzyl isothiocyanate. These results support the research outcome from other studies [48].

The food industry is very interested in finding natural antibiotic agents to reduce food spoilage caused by spore-forming bacteria. The toxins of some species can cause food poisoning. In addition, some of these pathogens often exhibit increased stability even at low temperatures or pH levels, making common sterilization methods difficult and not always possible if one wants to preserve the nutritional value and odor of the food. Therefore, a study was conducted to investigate the effect of *D. gossweileri* bark oil on the formation of *Bacillus* spores. Three different *Bacillus* species were used for this purpose (*Bacillus cereus* T, *B. subtilis* NCTC, *B. megaterium* 8174). Inhibition of spore germination was demonstrated by a macrodilution assay. MICs of the EO were determined for *B. cereus* T at 9 µg/mL, *B. subtilis* NCTC at 9 µg/mL, *B. megaterium* 8174 at 4 µg/mL. Further, the MICg (minimal concentration inhibiting germination of spores) of the three species were at 2 µg/mL, 1 µg/mL and 2 µg/mL, respectively. The EO also inhibited the germination and vegetative growth of spores of *Geobacillus stearothermophilus*. The MIC for this bacterium was 9 µg/mL. *D. gossweileri* therefore showed a stronger antimicrobial activity compared to other EOs in the study, such as *T. vulgaris* or *Zingiber officinale* Roscoe (*Zingiberaceae*). Various combinations of the EO of *D. gossweileri* with other EOs, like *Syzygium aromaticum* (L.) Merr. and L.M. Perry (*Mytraceae*) EO, were able to enhance antibacterial activity, which the authors attributed to the synergistic effect of benzyl isocyanate and other active compounds. Phenylpropanoids thymol and eugenol, as well as terpenoid hydrocarbons *trans*-β-caryophyllene and neral found in the combined EOs, seemed to support the transport of benzyl isocyanate into the cell [15].

Concentrations of 0.25, 0.5, and 1 µg/mL of *D. gossweileri* bark EO in orange juice and milk demonstrated increased sensitivity of spores to radiation sterilization procedures, which could be valuable to the food industry in the future. The authors suggested that the increased UV sensitivity to spores was due to the photosensitizing effect of benzyl isothiocyanate. The increased reactivity of this constituent could lead to binding to DNA and, with increased energy absorption trough exposure of UV, increased breakage and ROS formation. However, the characteristics of the food used also influences the sensitivity of a bacterium to EO. A stronger anti-spore effect was observed in milk than in orange juice. This could be due to the different pH values (orange juice: 4.65 versus milk: 6.75). A low pH increases the hydrophobicity of EO, thus facilitating access to the germinant receptors in the spore membranes. The high content of protein and high lipid concentration in the milk probably also plays a role in the mode of action. Nevertheless, it should be noted that the high concentration of isothiocyanates, which have a particularly strong pungent odor and taste, may be undesirable in food. However, in this study, only a slight change in the taste or odor of the orange juice was observed at 2.5 µg/mL, but a significant change was noted at 10 µg/mL [16].

The bark EO of *D. gossweileri* was further investigated for its antioxidant and antiinflammatory properties by Ndoye Foe et al. [18]. For this purpose, a DPPH radical scavenging assay was performed and the EO showed an antioxidant activity index (AAI) of 12.821 (scavenging concentration (μg/mL) was 0.20), which was ten times higher than that of ascorbic acid (AAI = 1.262 and SC_50_ of 1.98 μg/mL (SC_50_ = concentration of sample required to scavenge 50 % of free radicals)). According to the criteria of Scherer and Godoy AAI > 2 means that it has a very strong antioxidant activity [49]. The authors concluded that the antioxidant properties probably were caused by synergetic effects between different components in the oil. The ferric reducing capacity of *D. gossweilei* EO was reported to be 0.76 ± 0.03 μg AAE/mg (ascorbic acid equivalent per mg EO) and the total phenolic content was 365.38 ± 0.66 µg AAE/mg. Anti-inflammatory activity was determined by bovine serum albumin anti-denaturation assay. *D. gossweileri* bark EO showed an IC_50_ value (μg/mL) of 88.30 compared to a 104.44 μg/mL IC_50_ value of diclofenac sodium [18].

Another study showed antimycobacterial activity of *D. gossweileri* bark EO against virulent strain H37Rv and two isolates IS53 and IS310 of *Mycobacterium tuberculosis*. The EO exhibited cell membrane lysis and DNA inhibition. Disruption of cellular homeostasis by structural changes in the membrane and cell wall caused bacterial death. The MIC values (µg/mL) against H37Rv, IS53 and IS310 were 4.88, 9.76 and 4.88, respectively. In contrast, no mycobacterial activity was detected against the multidrug resistant and extensive drug resistant MJ and UJ isolates and the isoniazid resistant isolate AC79. Against the rifampicin resistant isolate AC45, the bark EO showed moderate activity with a MIC of 156.25 µg/mL [17].

### 2.4. Cryptocarya massoy (Oken) Kosterm. (Lauraceae)

*C. massoy*., also known as *C. aromatica* (Becc.) Kosterm. or *Massoia aromatica* (Becc.), is an evergreen tree found in Papua New Guinea and Indonesia [21]. It grows up to 15–30 m tall. The bark is characterized by its smooth, greenish, or light grey appearance. It has a thickness of 5–15 mm [50]. The EO ingredients from the wood and especially the bark of the tree are popular in perfumery. The oil is also said to have antifungal and insect repellent properties. In addition, it is used as a natural coconut flavoring in food products. Since the aroma can also be perceived as butter-, cream- or milk-like, it is also used in the food industry as milk aroma [21,23,51].

Traditional uses of the bark include stomach cramps during pregnancy, post-pregnancy recovery, and fever. It further is applied as a tranquillizer/sedative, to stimulate sex hormones, as a tonic, as an antispasmodic, and as an anthelmintic [50]. It is part of an indigenous herbal medicine from Indonesia called “pilis” to restore vitality and stimulate recovery after childbirth [52].

An analysis of the EO of the bark identified over 20 constituents. Of the 98.61% identified oil massoia lactones were the main components of the EO. This term describes α, β-unsaturated δ-lactones. Some of them could be chemically identified: C-10 massoia lactone (5,6-Dihydro6-pentylpyran-2-one; Figure 1j) with a 56.25% relative content, C-12 massoia lactone (5,6- Dihydro-6-heptylpyran-2-one) with 16.51%, C-8 massoia lactone (5,6-Dihydro-6-propylpyran2-one) with 3.4% and C-14 massoia lactone (5,6-Dihydro-6-nonylpyran-2-one) with 0.56%. Additionally, the saturated compounds δ-decalactone (1.53%), δ-dodecalactone (0.49%), tuberolactone (0.67%) were identified. Benzyl benzoate (12.72%) and benzyl salycilate (1.79%) as aromatic esters are also part of the EO. It is very interesting that common dimethylallyl pyrophosphate or phenylpropanoid pathway derivatives were hardly detected in the EO of the bark. No typical monoterpenes and sesquiterpenes are found, which distinguishes this EO from others. The entire *Cryptocarya* genus has very unique secondary metabolites [21]. A comparative study of the bark and heartwood oils again revealed C-10 massoia lactone as the main constituent of the bark EO, as well as C-12 massoia lactone and benzyl benzoate. In contrast to the previous analysis, a very small amount of linalool, borneol and β-bisabolene were found in the bark oil. The heartwood oil also contained C-10, C-12, and C-14 massoia lactones and δ-decalactone [53].

In addition to *Cryptocarya*, massoia lactones can also be found in *Aeollanthus suaveolens* Mart. ex Spreng. (*Lamiaceae*), *Baccharis magellanica* Pers. (*Asteraceae*), *Isolona cooperi* Hutch. and Dalziel ex G.P.Cooper and Record (*Annonaceae*), and *Cenchrus macrounrus* (Trin.) Morrone (cattail balm; *Poaceae*). In addition, they are part of the defense substances of two ant species (genus *Camponotus*) [21], C-10 massoia lactone being the most important one in the ants’ defense mechanism [53].

In particular, the massoia lactones and benzyl esters in the EO have shown phototoxic properties against the germination and growth of *Lycopersicon esculentum* Mill. (*Solanaceae*) and *Cucumis sativus* L. (*Cucurbitaceae*) plantlets, which were used in a study as representatives of typical target plants for herbicides. They were equivalent to or better than EO in commercial weed and pest control products [21,51].

As for antifungal activity, *C. massoy* EO showed potential as an anti-biofilm compound. *C. albicans* causes many opportunistic infections, especially in people with compromised immune systems. It can form biofilms that are very difficult to treat because they can also grow on medical devices such as implants, catheters, and ventilators. Therefore, the way of action the EO affects the formation and degradation of biofilms was investigated. It was found that *C. massoy* bark EO showed synergistic effects with several other EOs, such as *Cinnamomum burmanii* (Nees and T. Nees) Blume (*Lauraceae*) and *Ocimum basilicum* L. (*Lamiaceae*), in the intermediate phase of *C. albicans* biofilm. Further synergistic effects with the EO of *C. burmanii* and *Citrus hystrix* D.C. (*Rutaceae*) in the mature phase were reported [19]. In another study, anti-biofilm activity, specifically of the C-10 massoia lactone, against *C. albicans* showed an IC_50_ of 0.026 µg/mL [20]. The antimicrobial activity of the EO of the bark was tested against four phytopathogenic bacteria. *Agrobacterium tumefaciens*, *A. vitis*, *Clavibacter michiganensis* and *Pseudomonas syringae*. The EO showed MIC (µL/L) values of 250, 2000, 250 and 125, respectively. *S. aromaticum* EO was used as reference and showed MIC (µL/L) values of 250, 2000, 1000 and 500, respectively [21]. Breakpoints of MIC values for EOs have not been defined so far, impeding the interpretation of reported antibacterial activities. Comparing drugs with EOs, a MIC value of 2000 is rather high and therefore indicates low to no activity. In our opinion it could be concluded that *C. massoy* EO showed a promising effect against *A. tumefaciens*, *C. michiganensis* and *P. syringae*, but was inactive against *A. vitis*. *S. aromaticum* indicated a good effect against *A. tumefaciens* and a fair antimicrobial activity against *C. michiganensis* and *P. syringae*. However, another problem in the evaluation of antimicrobial activity of EOs is the fact, that research groups use different methods to determine antimicrobial activity, making a comparison of results impossible. For future research in this important field it would be necessary to create and use standardized methods. The immunomodulatory effect of *C. massoy* EO was demonstrated using an in vitro mouse macrophage phagocyte assay. The response of macrophages to latex beads and *C. albicans* was observed. The oil and lactone successfully increased the phagocytosis activity of macrophages and inhibited *C. albicans*. In an in vitro lymphocyte proliferation assay, no proliferative effect on lymphocytes was observed. A Cytotoxicity assay with the bark EO and the isolated C-10 massoia lactone revealed concentration dependent cytotoxicity to the test cell lines. Vero and primary culture of fibroblast cells were used. In fibroblast cells, the IC_50_ of C10 massoia lactone and EO were 11.29 μg/mL and 26.07 μg/mL, respectively. The IC_50_ values of Vero cells were 28.35 μg/mL (C-10 massoia lactone) and 37.34 μg/mL (EO), respectively. The study results indicated that massoia oil and its isolated main compound C-10 massoia lactone seems to have the capability as an immunmodulator [20].

The α,β-unsaturated δ-lactone structure, which is also part of massoia lactones, plays a central role in various studies on anti-inflammatory and immunomodulatory properties. They are considered to be cytotoxic. In particular, C-10 massoia lactone is thought to be responsible for cytotoxic effects on different cancer cell lines [20,22]. This structural part is found in various components of plants and marine organisms and many of them have specific biological activities. Therefore, because of its particularly simple structure, further reports on the (-)-massoia lactone from the bark of *C. massoy* may provide more information on future syntheses of various complex natural products that also include this core functional group. Further modification of the core structure may lead to the preservation of various biological properties of already known complex natural products [22].

Batubara et al. studied the effects of *C. massoy* EO and its major constituent massoia lactones after inhalation in Dawley rats in vivo. The animals that inhaled the fraction of EO rich in massoia lactones reduced their body weight due to the appetite suppressant effect it showed. It also prevented fat deposition in liver cells. It further was found that the fractionated EO had a limiting effect on triglycerides and cholesterol concentration in the blood [23].

The fear that *C. massoy* could be overharvested because of its many uses as a medicine and in food industry is pervasive. In Indonesia, *C. massoy* EO is the eighth most produced EO. However, studies show that it is possible to grow *C. massoy* outside its natural habitat [54].

### 2.5. Vanillosmopsis arborea Barker (Asteraceae)

*V. arborea*, also known as “Candeeiro”, is a tree native to Brazil, especially in the Caatinga biome in the northeast. It plays an important commercial role due to the ingredients that can be extracted from it [24]. Many studies on the bark EO have shown larvicidal, antileishmanial, antimicrobial, antimicrobial, anti-inflammatory, antinociceptive, and gastric protective activities. More cultural uses are documented of this tree, for example it is said to repel mosquitoes. It is also discussed to use the EO as an adjuvant in the treatment of respiratory bacterial infections in combination with antibiotics [24]. The analysis of the EO of the bark of *V. arborea* by Teixeira et al. showed that α-bisabolol (70%), α-cadinol (8.4%), elemicine (6.21%), β-bisabolene (4.46%), δ-guaiene (2.31%), β-cubebene (1.76%) and estragole (1.08%) were the main parts of its chemical composition [24].

Due to environmental conditions, which—besides others—include age, water availability or seasonality, there are differences in the proportions of the main components in different EO analyses [24]. This is also evident from further photochemical analysis in other studies with the components (-)-α-bisabolol (97.9%), methyl eugenol (1.6%), and bisabolol oxide (0.5%) [29]. Another investigation performed by Santos et al. [25] again confirmed α-bisabolol as the main constituent of the stem oil with 80.47%. Other components were ethyl propanoate (5.87%), propyl ethanoate (9.00%), o-methyleugenol (2.39%) and bisabolol oxide (2.31%). 24 Sesquiterpenoids such as α-bisabolol, found in high concentrations in EO, are also often associated with pheromone, insect repellent, or phytoalexin properties. α-Bisabolol plays an important role in cosmetics due to its skin-healing properties. As mentioned earlier, low toxicity and anti-inflammatory activity are the properties that make these compounds so attractive for further research and wide use in medicine. Efforts are being made to use fewer and fewer antibiotics to counteract the development of resistance. The mechanism of action of EOs is based on the destabilization of the bacterial cell wall to achieve better absorption of the antibiotic. The particularly hydrophobic property of the EO plays a major role in this process. Tests indicated a synergistic effect of the EO of *V. aroborea* with the antibiotic gentamycin on the strain of *P. aeruginosa*. At the same time, in the same experiment, the effect of EO together with tetracycline and tobramycin did not show increased efficacy of the respective antibiotic. Therefore, *V. aroborea* EO seems to have potential as an enhancer in antibiotic therapy [25].

An animal experiment in which oral doses of 200 mg/kg and 400 mg/kg of bark EO were administered *per os* to male albino mice, was performed by De O Leite and coworkers [28]. Results showed a significant effect of the oil on ethanol-induced gastric lesions in the experimental animals. The authors concluded from the series of experiments that an agonism of the peripheral α2-receptor is likely to be considered as part of the mechanism of gastroprotection [28]. Peripheral antinociceptive activity and an antipruritic effect were demonstrated by Leite et al. with various tests in male mice [29]. Other animal studies described the analgesic and anti-inflammatory effects of the EO. A sedative effect has also been demonstrated [30]. The antinociceptive effect of α-bisabolol-rich EO has been discovered to derive from downregulation of neuronal excitability and the inhibition of inflammatory mediators and cytokines [55]. Target for bisabolol is probably the TRPA1 channel which plays a major role in the release of TNFα, which is one of the most important inflammatory mediators in the human body. Studies also showed a higher IL-10 expression, which is known for its anti-inflammatory effect [56]. Furthermore, the downregulation of the FOS protein in the spinal cord through an EO-β cyclodextrin-complex proved to be a promising treatment option for orofacial pain [55].

In another study, using the branches of the *V. arbora* tree for distillation and subsequent GC-MS analysis, a concentration of 91.02% α-bisabolol was determined. This main compound, a sesquiterpene alcohol, plays an important role in the mode of action of EO. Tests of EO and α-bisabolol on three different strains of *C. albicans* (CA INCQS 40006, CK INCQS 40095, CT INCQS 40042) resulted in IC_50_ (μg/mL) values of 9.88, 23.89 and 584.9 for EO and 8.92, 18.15 and 518.26, respectively. There were no significant differences between the EO and the isolated compound. To test the behavior on bacterial strains, four antibiotics (amikacin, gentamicin, ampicillin, and benzylpenicillin) were tested on two multidrug-resistant pathogens (*S. aureus* 03 and *E. coli* 08) along with the entire EO and the isolated α-bisabolol. In all tests, a synergistic effect was observed between the antibiotics and the EO as well as the isolated α-bisabolol. This could open many doors in the treatment of resistant bacterial strands. Combination with conventional antibiotics could result in lower toxicity, lower minimum effective dose, and fewer side effects during antibiotic therapy. Again, α-bisabolol is likely responsible for the antibiotic-modulating effect [26]. Further studies on EO of V. arbora and α-bisabolol showed in toxicity against *Leishmania amazonensis* in vitro. These investigations determined IC_50_ values of 7.35 µg/mL and 4.95 µg/mL of EO and α-bisabolol against promastigotes and IC_50_ values of 12.58 µg/mL and 10.70 µg/mL against intracellular amastigotes, respectively, without toxic activity against the treated macrophages. The authors mentioned that the effect on the lipid metabolism of the parasites led to the accumulation of lipid precursors and caused morphological changes in the cell membrane, resulting in the antileishmanial activity. This indicates further development and future therapeutic use [27].

α-Bisabolol is commercially extracted from the EO of the tree and has been extensively studied. In human and rat malignant glioma cells, it caused cell death by inducing apoptosis. Cavalieri et al. described an involvement of the mitochondrial permeability transition. This is associated with the translocation of cytochrome C, suggesting that the whole process triggers the intrinsic pathway of apoptosis. This process of cytotoxicity on glioma cells has been shown to be time and dose dependent. However, the advantage of this compound is at the same time its low toxic effect on animals, as shown in an animal experiment with a rat administered 120 mg/kg α-bisabolol, which showed no toxic effects after 24 h [57]. The low toxicity is also a reason why α-bisabolol is popular in cosmetics and perfumes [27]. These results underscore the potential inherent in this compound. With a CL_50_ of 15.9 mg/mL and CL_90_ of 28.5 mg/mL the EO of *V. arborea* further showed very high larvicidal activity [58].

### 2.6. Cedrus deodara (Roxb. Ex. D. Don) G. Don (Pinaceae)

*C. deodara* is also known as “Himalayan cedar”. This tree is found on the slopes of the Himalayas at an altitude of about 1650 to 2400 m above sea level. This includes regions in northern and central India, northern Pakistan, western Nepal, eastern Afghanistan and southwestern Tibet. The evergreen tree can reach a height of 65–85 m and a diameter of 4 m. The bark *of C. deodara* is dark grayish-brown or reddish-brown in color and has diagonal and vertical cracks [33,59]. The EO of the wood, as well as of the bark is used in medicine. Antiseptic, insecticidal, antifungal, and anti-inflammatory are just some of the properties attributed to the EO. It is also said to be applied against various animal diseases and as a molluscicide. Commercially, the EO is an important ingredient for the perfume industry and cosmetics, as well as in soap making [59]. Especially EOs of the *Pinaceae* family are known for their antibacterial, larvicidal and antioxidant activities [60,61].

Reports of traditional Indian uses of *C. deodara* wood oil and bark powder include application against colds, coughs, cancer, skin diseases, bronchitis, diarrhea, tuberculosis, urination, itching, hemorrhoids, mental disorders, and blood diseases. In addition, the vapors of the bark have been used as a snake repellent, but also against scabies and external parasites. The bark and the wood are the most used parts of *C. deodara* [33].

A characterization of the yellow-colored EO of the bark identified by GC-MS described monoterpenes (44.67%), sesquiterpenes (36.43%), and diterpenes (18.88%). The largest proportions were listed as α-pinene (24.28%), *cis*-ocimene (14.11%; Figure 1k), longifolene (8.28%; Figure 1l), cembrene (6.69%), α-copaene (6.56%), α-terpineol (3.73%), and verticellol (3.03%) [31].

Subsequent tests against three fungal pathogens *Curvularia lunata* (IC_50_ value of 2.22 ± 0.13 µL/mL), *Alternaria alternata* (IC_50_ of 3.71 ± 0.12 µL/mL) and *Bipolaris spicifer* (IC_50_ of 4.8 ± 0.34 µL/mL) showed antifungal activity, which is probably caused by the high content of oxygenated monoterpenes in the EO of the bark. Further investigation of the antifungal effect on *C. lunata* identified a mixture of ten different components as the cause. This suggests that a synergistic effect seems to play an important role in the mode of action. Molecular docking studies were performed that analyzed the binding affinity of the antifungal fraction of *C. deodara* bark oil to the protein tri-hydro-naphthalene reductase (ID-1YBV). Inhibition of pentaketidal melanin biosynthesis was determined to be the antifungal mechanism of action. This makes the bark EO a potential candidate fungicide against *C. lunata* [31]. α-Pinene, a major compound in the EO of the bark, showed its antimicrobial properties against *E. coli, S. aureus*, *Micrococcus luteus*, and *B. subtilis* [32].

Analysis of the wood EO indicated more than 34 constituents. Mainly α-himachalene (17.1%; Figure 1m) and β-himachalene (38.8%; Figure 1n) which are sesquiterpenehydrocarbons with a hexahydrobenzocycloheptene basic skeleton and are not found in the bark oil. Cedarwood oil with a high proportion of himachalenes is called “super rectified oil” and with a high proportion of atlantones it is called “perfume quality” [61].

Other compounds found in wood EO by GC analysis were γ-himachalene (12.6%; Figure 1o), (E)-α-atlantone (8.6%), *cis*-α-bisabolene (2.2%), (E)-γ-atlantone (2.4%), (Z)-γ-atlantone (2.3%), and (Z)-α-atlantone (1.4%). With an overall identification of 98.3%, this accounts for 79.5% sesquiterpene hydrocarbons and 18.9% oxygenated sesquiterpenes [61]. In contrast, Tripathi et al. found atlantones and himachalene epoxides to be major constituents in their study of cedarwood oil [62].

Several studies have demonstrated the analgesic and anti-inflammatory effects of wood EO in chronic and acute stages. One main cause is the inhibition of the release of inflammatory prostaglandins through the inhibition of cyclooxygenase enzymes. Other studies also supported the membrane stabilizing effect of the oil. More research needs to be done to investigate exactly which molecules cause the underlying mechanisms. In this anti-inflammatory context, stabilization of mast cells was observed in rats, which could be part of the anti-asthmatic mechanism. There are too few studies to fully explain this [33].

In addition, studies have looked at the insecticidal property. Fractions containing himachalene as the main component showed particularly pronounced insecticidal properties against *Musca domestica* (house fly) and *Callosobruchus chinensis* (pulse beetle). Further development of these components could provide a potent insecticide [33].

Finally, the toxicity of the wood oil was studied in mice. Animals receiving a dose of 200–400 mg/kg showed signs of depression and mice receiving 500 mg/kg of the oil showed 50% mortality [33]. Its traditional use as a remedy against gastric ulcers was further investigated in another animal experiment on Wistar rats. *C. deodara* EO at a dose of 100 mg/kg successfully reduced the number of ulcers, ulcer score and ulcer index. It indicated inhibition of pyloric ligation induced gastric ulcers with 41.49%. Standard drug rabeprazole, used as control in the investigation, inhibited 67.74% at 20 mg/kg. Ethanol-induced ulcers showed inhibition of 50.17% at a dose of 100 mg/kg, while rabeprazole 20 mg/kg showed inhibition of 69.95% of ulcers. The exact mode of action remains to be investigated [34]. Against K562 cancer cell lines, *C. deodara* wood oils revealed an IC_50_ of 37.79 μg/mL. According to other investigations, reviewed by Kumar and coworkers, the active compounds causing the cytotoxic effect are believed to be α-pinene, β-pinene, caryophyllene, and terpineol. β-Pinene showed antiproliferative properties against MCF-7 breast cancer, A375 lung cancer and HepG2 liver cancer cells in various studies [32]. Recent studies on a virulence agent of SARS-CoV-2 investigated molecules containing the backbone of α,β,γ-himachalenes from *C. deodara*. The goal was to find a potential inhibitor of nonstructural protein 1 that restricts viral replication. The reason was that himachalene scaffolds are known for their anti-inflammatory and antiviral properties [63].

## 3. Conclusions

This overview is intended to illustrate the different biological properties and chemical composition of the various bark oils. Ultimately, however, it is also important to put the commercial benefits into a realistic perspective. It must be clarified whether the plants can be grown sustainably on plantations and thus used on a large scale. Soil conditions also affect the essential oil, which will certainly present some challenges in the future. From a European perspective, the bark oils seem to come from trees that are rather uncommon or exotic. One reason for this could be that in Europe the trees are cut down producing EO from wood and bark while in certain other areas of the world the outer bark is scraped off and used independently from the wood. The centuries-long use of bark for clothing, building materials, or simply as traditional medicine provided a great deal of traditional knowledge. The easier accessibility of bark plays an important role. Due to the many documented applications of bark oils [64], they will continue to gain importance in various research areas in the future.

The antimicrobial properties of some bark oils mentioned, such as *V. arborea*, are particularly noteworthy. The World Health Organization’s global report on surveillance on antimicrobial resistance has highlighted that emerging resistance is a global threat. The treatment of multidrug-resistant pathogens is a major challenge for medicine [65]. Reasons for this include the high use of antibiotics and inefficient or incorrect use. It is also important to consider current world events. Approximately 15% of hospitalized COVID-19 patients develop secondary bacterial infections such as bacterial pneumonia or require antibiotics for prophylaxis because they are on a ventilator [66]. This again highlights the need for alternatives and new solutions. One approach could be to focus on plants with constituents that have antibiotic potential. EOs can influence the mode of action of antibiotic substances. Studies have shown that the oil from the bark of *V. arborea* had an enhancing effect when combined with various antibiotics. Exactly such synergistic enhancing effects with antibiotics have also been observed with cinnamon oil and its main constituent [67]. This supportive effect could be used in the future to circumvent the resistance of multidrug-resistant pathogens and reduce the use of antibiotics. Thus, they could also help to reduce the high incidence of ever-increasing antibiotic resistance. Lowering the dose of antibiotics by combining them with an adjuvant such as EOs could also reduce side effects by lowering the minimum effective dose of the drugs.

However, most of the studies discussed in this review were performed in vitro (Table 1) and—although their results give a direction for future research—their outcome may not be directly transferable to a clinical setup. Therefore, clinical studies as well as applied investigations using EOs have to be performed and supported. Recently, for example, Hawkins and coworkers determined the energy-level-increasing effect of various EOs in women with post-COVID-19. In a randomized double bind trial, patients inhaled blends of EOs for two weeks. Compared to the control group, fatigue level significantly decreased (*p* = 0.02) [68]. In another study *Lavandula angustifolia* L. (*Lamiaciae*) EO showed an antimicrobial effect especially against *Staphylococcus* when being dispensed over two months in hospital environments [69]. A further investigation determined a significant reduction of microbial contamination in a hospital by vaporizing a mixture of lemon and fir EO. That way, concentrations of bacteria and fungi in the air was reduced by around 40% after two hours [70]. Nevertheless, in vivo studies further need to be conducted in the future to provide a more accurate picture of the toxicity of certain antimicrobial bark EOs on the human body and additionally their effect on the human microbiome. The routes of administration also need to be clarified, as well as pharmacodynamic and pharmacokinetic properties, which will certainly be a challenge for the future. More studies need to be done to get a better understanding of the mechanism of action of bark EOs and their components. Additionally, studies should focus on the efficacy and use of various bark oils, which have not yet been extensively researched.

## Figures and Tables

**Figure 1 molecules-27-07295-f001:**
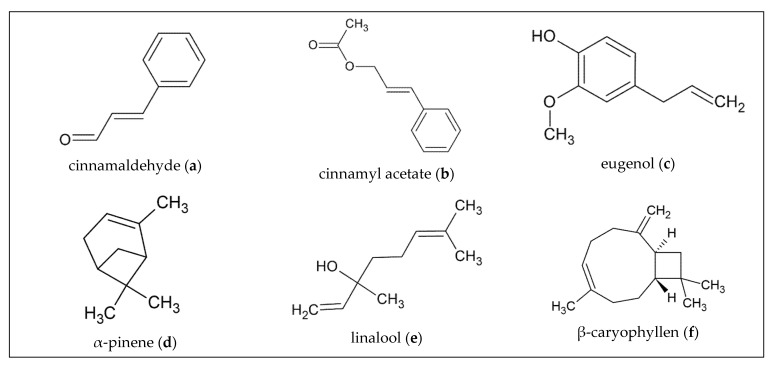
Chemical structures of principal constituents of selected barks and wood.

**Table 1 molecules-27-07295-t001:** Major biological activities of selected bark EOs.

Essential Bark Oil (Plant Family)	Biological Activity (Model)	References
*Cinnamomum verum* J. Presl (*Lauraceae*)	anticoagulant (in vitro) antimicrobial (in vitro) antidiabetic (in vitro) anti-Alzheimer (in vitro)	[7] [8] [9] [10]
	antioxidant (in vitro)	[11]
	neuroprotective (in vitro) anti-cancer (in vitro) anti-inflammatory (in vitro)	[12] [8] [13]
*Cedrelopsis grevei* Baill. (*Ptaeroxylaceae*)	antioxidant (in vitro) antibacterial (in vitro)	[14]
	anti-cancer (in vitro)	
*Drypetes gossweileri* S. Moore (*Euphorbiaceae*)	antibacterial (in vitro)	[15,16,17]
	antioxidant (in vitro)	[18]
	anti-inflammatory (in vitro)	[18]
*Cryptocarya massoy* (Oken) Kosterm. (*Lauraceae*)	anti-biofilm (in vitro) antibacterial (in vitro)	[19,20] [21]
	immunomodulatory (in vitro) anti-inflammatory (in vitro) blood-fat-reducing (in vivo)	[20] [20,22] [23]
*Vanillosmopsis arborea* Barker (*Asteraceae*)	antibacterial/antibiotic-synergistic (in vitro) gastroprotective (in vivo) antinociceptive (in vivo) anti-inflammatory (in vitro)	[24,25,26,27] [28] [29] [30]
*Cedrus deodara* (Roxb. Ex. D. Don) G. Don (*Pinaceae*)	antifungal (in vitro) antibacterial (in vitro) insecticidal (in vitro) gastro-protective/anti-ulcer (in vitro)	[31] [32] [33] [34]

## Data Availability

Not applicable.

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
