# Peer review of "Beyond the Bark: An Overview of the Chemistry and Biological Activities of Selected Bark Essential Oilsâ€"

_molecules, 2022, doi:10.3390/molecules27217295_

Round 1
Reviewer 1 Report
This is a well written manuscript. English and style are good. The work is innovative and attempts to improve the acknoweledge about brk essential oils. I have a number of minor recommendations:
Line 178 hydrodistillation instead of hido….
Line 223 In the title of each paragraph with the name of the plant, it would be more appropriate to write the full name of the plant with the name of the family to which it belongs. In this way, the reader immediately understands what plant it is. In the description of the plant, write the abbreviated name. Line 167 is an example. 2.2 Cedrelopsis grevei Baill. (Ptaeroxylaceae). Line 168 C.grevei is a tree native.....
Please change each title paragraph 2.3, 2.4, 2.5. etc.
Line 282 and the following. I agree to show MIC values; however, authors should comment on what these values mean. Since breakpoints for essential oils are not available, a MIC value should be accompanied with a comment on the activity of the oil (i.e., good activity, low activity, etc.). If we compare drugs and oils, a MIC value of over 3,000 mcg/ml is extremely high and it could not be associated with good activity (lines 309-310).. If we would like to evaluate the antimicrobial activity of the essential oils from a clinical point of view and future use, we have to use methods that are as standardized as possible in order to conform to those used for the evaluation of antibiotic antimicrobial activity. If each of us used our own method, the results would be speculative, making it difficult to compare them with other investigators and determine whether the product is valid. As a control, it is also necessary to use an antibiotic.
Please add a comment for each plant regarding MICs and antimicrobial activity.
Line 373 please add the family name of the plants (Lycopersicum and Cucumis)
Line 434 and the following. Please specify if the components are related to the activity of this oil and which activity.
Line 438 Santos et al. [46] (add the reference number here after the author names or at the end of the sentence (….stem oil with 80.47%).
Line 455 Please add reference De O Leite and coworkers [47].
Lines 545-550. Please specify if the components are related to the activity of this oil and which activity.
Line 560 Please write the Latin name in full if it is the first time they are mentioned.
Author Response
Dear reviewer,
thank you very much for your suggestions to improve our manuscript. please find below our response:
Line 178 hydrodistillation instead of hido CORRECTED
Line 223 In the title of each paragraph with the name of the plant, it would be more appropriate to write the full name of the plant with the name of the family to which it belongs. In this way, the reader immediately understands what plant it is. In the description of the plant, write the abbreviated name. Line 167 is an example. 2.2 Cedrelopsis grevei Baill. (Ptaeroxylaceae). Line 168 C.grevei is a tree native..... CORRECTED AS TO YOUR SUGGESTIONS
Line 282 and the following. I agree to show MIC values; however, authors should comment on what these values mean… CORRECTED, A PARAGRAPH ON THIS MATTER HAS BEEN ADDED, COMMENTS HAVE BEEN MADE.
Line 373 please add the family name of the plants (Lycopersicum and Cucumis) DONE
Line 438 Santos et al. [46] (add the reference number here after the author names or at the end of the sentence (….stem oil with 80.47%). DONE
Line 434 and the following. Please specify if the components are related to the activity of this oil and which activity. Lines 545-550. Please specify if the components are related to the activity of this oil and which activity. WE ACTUALLY DO NOT KNOW WHAT YOU MEAN. COMMENTS ON WHICH COMPOUND MIGHT BE RESPONSIBLE FOR THE ACTIVITY OF THE OILS WERE DONE IN THE CORRESPONDING PARAGRAPHS – NOT IN THE CHEMISTRY DESCRIPTION.
Line 455 Please add reference De O Leite and coworkers [47]. DONE
Line 560 Please write the Latin name in full if it is the first time they are mentioned. DONE
we hope we could correct our manuscript to your satisfaction and our manuscript can now be published in “Molecules”!
kind regards, Iris Stappen
Reviewer 2 Report
The present work is a review about essential oils extracted from barks of plant species with economical, medicinal, and ethnopharmaceutical values. Althought interesting, I have some suggestions for the authors to improve this collection of information.
1. Please, insert the patronymic to all scientific names of plant species reported in the text.
2. In the first paragraph major enphasis on the chemical components detectable in essential oils should be indicated (i.e., terpenes and their derivatives - biochemical pathways, classification, chemico-physical properties).
3. in the conclusions, new perspectives for the application of essential oils should be provided. For instance, I have found that recently a new se of this natural product/extract in hospital environments, as source of antineoplastic compounds, or as food preservative. It should be reported more in detail in the text as future perspecitves. For instance, mention the following papers: Journal of Herbal Medicine, 2021, 26: 100426; Journal of Environmental Science and Health, Part A, 2019, 54.3: 256-260; Food Chemistry, 2020, 330: 127268; Frontiers in microbiology, 2017, 7: 2161; Molecules, 2012, 17.3: 2704-2713; South African Journal of Botany, 2020, 133: 222-226.
4. A table or an image with the 6 plant species described in detail in the work and the list of their biological activity (already documented in literature) as a summary of the whole paper could be helpful for the reader.
Author Response
Dear reviewer,
thank you for your suggestions to improve our review! please find below our response:
- Please, insert the patronymic to all scientific names of plant species reported in the text. DONE
- In the first paragraph major enphasis on the chemical components detectable in essential oils should be indicated (i.e., terpenes and their derivatives - biochemical pathways, classification, chemico-physical properties). DONE, A PARAGRAPH WAS ADDED
- in the conclusions, new perspectives for the application of essential oils should be provided. For instance, I have found that recently a new se of this natural product/extract in hospital environments, as source of antineoplastic compounds, or as food preservative. It should be reported more in detail in the text as future perspecitves. For instance, mention the following papers: Journal of Herbal Medicine, 2021, 26: 100426; Journal of Environmental Science and Health, Part A, 2019, 54.3: 256-260; Food Chemistry, 2020, 330: 127268; Frontiers in microbiology, 2017, 7: 2161; Molecules, 2012, 17.3: 2704-2713; South African Journal of Botany, 2020, 133: 222-226.
A paragraph was added at in the “Conclusion”. however, we did not insert all above suggested literature, because 2017 and 2012 did not seem us to be recent findings. Food chemistry 2020 is a review that we included in the introduction on EO chemistry (suggested by reviewer 1)
- A table or an image with the 6 plant species described in detail in the work and the list of their biological activity (already documented in literature) as a summary of the whole paper could be helpful for the reader.
We created a table, as suggested. However, since this review is on bark essential oils, a picture of the tree does not seem us to be improving.
we hope that we could correct our manuscript as to your satisfaction and it therefore can be published in Molecules!
kind regards, Iris Stappen